# Supramolecular Amphiphiles Based on Pillar[5]arene and Meroterpenoids: Synthesis, Self-Association and Interaction with Floxuridine

**DOI:** 10.3390/ijms22157950

**Published:** 2021-07-26

**Authors:** Alan A. Akhmedov, Dmitriy N. Shurpik, Pavel L. Padnya, Alena I. Khadieva, Rustem R. Gamirov, Yulia V. Panina, Asiya F. Gazizova, Denis Yu. Grishaev, Vitaliy V. Plemenkov, Ivan I. Stoikov

**Affiliations:** 1A.M. Butlerov Chemical Institute, Kazan Federal University, 18 Kremlevskaya Street, 420008 Kazan, Russia; naive2294@gmail.com (A.A.A.); padnya.ksu@gmail.com (P.L.P.); as-alex93@mail.ru (A.I.K.); gamirov_21@mail.ru (R.R.G.); panintim@rambler.ru (Y.V.P.); Asiya_gazizova@mail.ru (A.F.G.); plem-kant@yandex.ru (V.V.P.); 2Scientific and Educational Center of Pharmaceutics, Kazan Federal University, 420008 Kazan, Russia; dionis.grishaev@yandex.ru

**Keywords:** terpenoids, macrocyclic systems, pillar[5]arenes, associations, host-guest systems, floxuridine

## Abstract

In recent years, meroterpenoids have found wide biomedical application due to their synthetic availability, low toxicity, and biocompatibility. However, these compounds are not used in targeted drug delivery systems due to their high affinity for cell membranes, both healthy and in cancer cells. Using the approach of creating supramolecular amphiphiles, we have developed self-assembling systems based on water-soluble pillar[5]arene and synthetic meroterpenoids containing geraniol, myrtenol, farnesol, and phytol fragments. The resulting systems can be used as universal drug delivery systems. It was shown by turbidimetry that the obtained pillar[5]arene/synthetic meroterpenoid systems do not interact with the model cell membrane at pH = 7.4, but the associates are destroyed at pH = 4.1. In this case, the synthetic meroterpenoid is incorporated into the lipid bilayer of the model membrane. The characteristics of supramolecular self-assembly, association constants and stoichiometry of the most stable pillar[5]arene/synthetic meroterpenoid complexes were established by UV-vis spectroscopy and dynamic light scattering (DLS). It was shown that supramolecular amphiphiles based on pillar[5]arene/synthetic meroterpenoid systems form monodisperse associates in a wide range of concentrations. The inclusion of the antitumor drug 5-fluoro-2′-deoxyuridine (floxuridine) into the structure of the supramolecular associate was demonstrated by DLS, 19F, 2D DOSY NMR spectroscopy.

## 1. Introduction

Nowadays researchers are paying more and more attention to natural compounds and their analogues [1,2]. Meroterpenoids are several separate classes of compounds resulting from mixed biosynthesis pathways. They occupy a special place among the diversity of natural structures [2,3,4]. Archaeal terpenoid lipids are one of the main classes of meroterpenoids. Archaea are one of the three domains of life [5,6,7]. Their cell membranes consist of a double layer of terpenoid lipids [8,9,10]. Archeosomes are a new generation of liposomes that consist of terpenoid lipids. In the last decade archeosomes have been widely used in odor preservation, food conservation, as well as for non-targeted drug delivery and prolongation of drug action [8,11]. The main difficulty of using archaeosomes in targeted delivery is their interaction with cells and excessively rapid release of the delivered substrate [8,11]. In this regard, the development of a new generation of archeosomes capable of providing targeted delivery of therapeutic agents is an interesting and worthwhile task.

A supraamphiphilic structure [12] based on meroterpenoids and a macrocyclic pillar[5]arene platform has been proposed to overcome the disadvantages of isoprenoid lipids. Pillararenes and well-known macrocyclic compounds such as crown ethers, cyclodextrins, calixarenes, etc. [13] have a tendency to form “host-guest” complexes [14]. At the same time they have a number of attractive characteristics, such as synthetic accessibility, planar chirality, a tubular spatial structure that forms an electron-donor cavity, and, as a consequence, the ability to accommodate fragments of “guests” in the macrocyclic ring [15]. Pillar[n]arenes tend to form complexes with positively charged molecules or molecules containing electron withdrawing groups [16]. Pillar[5]arene complexes with charged pyridinium [17] or imidazolium [18] salts can be built according to the “host-guest” principle. The presence of these properties opens up new opportunities in the formation of vesicles, transmembrane artificial channels, nanoreactors, metal-organic frameworks, liquid crystals, and supramolecular polymers [19].

In this article, we propose and develop a synthetic non-covalent self-assembling system of supramolecular amphiphiles based on “host-guest” complexes of pillar[5]arene and meroterpenoids as a universal drug delivery system (DDS).

## 2. Results and Discussion

### 2.1. Synthesis of Meroterpenoids Containing Pyridinium and Imidazolium Fragments Based on Terpene Alcohols

It is known [17,18,19,20,21] that compounds containing pyridinium or imidazolium fragments have a tendency to form ‘’host-guest’’ complexes with pillar[n]arenes. Therefore, we had been aimed to synthesize meroterpenoids containing pyridinium and imidazolium fragments. A supramolecular amphiphiles based on synthesized meroterpenoids and pillar[5]arene can be created by a “host-guest” interaction. The pillar[5]arene can act as a water-soluble component, and the meroterpenoid fragment can act as a lipophilic fragment.

Geraniol, myrtenol (monoterpenols), farnesol (sesquiterpenol) and phytol (diterpenol) were chosen as initial terpenoids for this task. These linear acyclic terpene alcohols have a structure similar to archaeal lipid polyprenols. These terpenols were chosen to create synthetic self-assembling supramolecular analogs of archeosomes on their basis.

At the first stage, we proposed to convert terpene alcohols into terpenyl bromoacetates to create highly reactive alkylating agents (Scheme 1). The compounds **2a,b** have been already described in the literature [22,23], however, methods of their preparation were rather complicated. Therefore, an original technique was developed. *N*,*N*-Diisopropylethylamine (DIPEA) was used as a base, chloroform as a solvent, the synthesis was carried out at a low (−5 °C) temperature. In this way, terpenyl bromoacetates **2a**–**d** were obtained with a good yields (Scheme 1).

The next stage was the synthesis of pyridinium and imidazolium salts based on terpene alcohols. Although compounds **3a,b** and **4b** were described earlier in the literature [23,24], we developed a more efficient method, which was also applicable to get derivatives of myrtenol and phytol. Diethyl ether was used as a solvent, and the synthesis was carried out at room temperature for 36 h. The products **3a,b,d** after isolation from the reaction mixture were analyzed by ^1^H-NMR spectroscopy. Target products **3a,b,d** and a small impurity of the starting terpenyl bromoacetate were obtained. The products were washed with hexane to remove impurities from the starting compounds. Only the initial phytyl bromoacetate **3c** was observed in the ^1^H-NMR spectrum after removal of the solvent in the case of the phytyl derivative. Apparently, the lipophilic phytyl fragment prevents the reaction due to the formation of aggregates of phytyl bromoacetate in diethyl ether. Then we tried to carry out the reaction of phytyl bromoacetate with pyridine in diethyl ether with heating under reflux. The target product **3c** was obtained using this approach.

The same method was used for obtaining imidazole-containing meroterpenoids **4a**–**d**. Terpenyl bromoacetates **2a**–**d** and methylimidazole were used as starting materials. Equimolar amounts of terpenyl bromoacetates and methylimidazole were used for the synthesis of imidazole-containing meroterpenoids (Scheme 1). Thus, it was possible to obtain geranyl and myrtenyl derivatives of methylimidazole. It was necessary to heat the reaction mixtures in the case of farnesyl and phytyl derivatives.

Thus, we have developed a method for the synthesis of the compounds **3a**–**d** and **4a**–**d**, based on the alkylation of pyridine and *N*-methylimidazole with terpenyl bromoacetates. The structure of the obtained meroterpenoids were confirmed by ^1^H-, ^13^C-NMR spectroscopy, FT-IR spectroscopy, and high-resolution mass spectrometry (see Appendix A).

### 2.2. Self-Assembly of Supramolecular Amphiphiles Based on Pillar[5]Arene and Meroterpenoids Containing Pyridinium and Imidazolium Fragments

Pillar[5]arenes have a tendency to form stable inclusion complexes in solutions with charged pyridinium and imidazolium fragments. In this regard, we proposed to obtain supramolecular amphiphiles in water. Decasubstituted pillar[5]arene **5** was chosen as a water-soluble component (Figure 1), which is a synthetically available model macrocycle [25]. Pyridinium and imidazolium salts **3a**–**d** and **4a**–**d** containing a terpenoid fragment were chosen as a lipophilic guest.

The observed upfield shift of proton signals of some fragments in the ^1^H-NMR spectrum indicates inclusion of this fragment into a pillar[5]arene cavity. We have studied the interaction of meroterpenoids **4a**–**d** with pillar[5]arene **5** in water by ^1^H-NMR spectroscopy, which showed that the charged fragment of imidazolium meroterpenoids **4a**–**d** is not included in the cavity of the pillar[5]arene, but rather ion replacement occurs. We decided to use methanol as a less polar solvent in order to lower the permittivity and change the type of ionic interaction between the macrocycle and meroterpenoid. As pillar[5]arene **5** is poorly soluble in methanol, it was decided to use a methanol/water mixture in a 2:1 ratio. Figure 2a shows a fragment of the ^1^H-NMR spectrum of mixture of the **4a/5** system (black) and meroterpenoid **4a** (gray) in a CD_3_OD/D_2_O (5 × 10^−3^ M) solution. The analysis of the obtained spectral results showed that the signals of the protons of the imidazolium fragment, methyl (CH_3_N) group, and CH_2_N^+^ fragment have an upfield shift. That data indicates the inclusion imidazolium fragments of meroterpenoid **4a** into the cavity of pillar[5]arene. Similar results are observed for myrtenyl derivative **4d** (Figure 2b) and farnesyl derivative **4b**. However, in the case of derivative **4c** in both D_2_O and CD_3_OD/D_2_O, the ion replacement occur in the presence of pillar[5]arene **5**. Apparently, this happens due to the higher lipophilicity of the phytol-based structure. Varying the solvents and temperature did not lead to the formation of the **4c/5** inclusion complex.

We have suggested that pyridinium salts, in contrast to imidazolium salts, are more prone to form “host-guest” complexes in water [26]. Figure 3a shows a fragment of the ^1^H-NMR spectrum in D_2_O (1 × 10^−3^ M) of the **3a/5** system (black) and meroterpenoid **3a** (gray). The signals of the proton of the pyridinium and CH_2_N^+^ fragment are shifted in the region of stronger fields by ~0.2 ppm. This data indicates the inclusion of the pyridinium fragment into the cavity of the pillar[5]arene **5** and the formation of an inclusion complex. Similar results (Figure 3b) are observed in the ^1^H-NMR spectrum of compound **3b** (a farnesol derivative) in the presence and absence of pillar[5]arene. Thus, pyridinium fragment of the synthesized meroterpenoids **3a–d** inclusion into the cavity of the pillar[5]arene **5** in water and form a supramolecular amphiphile according to the “host-guest” principle.

Further, systems **4a/5** in different rations of components in a methanol/water mixture (2:1) were studied by the dynamic light scattering (DLS) method (Table 1). It was shown that in the presence of pillar[5]arene **5**, the hydrodynamic particle diameter increases from 475 nm to 562 nm. Upon reaching a 5-fold excess of macrocycle **5** in a meroterpenoid/pillar[5]arene mixture, a sharp increase in the hydrodynamic diameter and polydispersity index (PDI) of aggregates occurs. This indicates coagulation of particles. Similar results were obtained when studying the interaction of imidazolium salts **4b** and **4d** with pillar[5]arene **5**. In the case of farnesyl and myrtenyl derivatives **4b,d**, particle coagulation begins at a meroterpenoid/pillar[5]arene ratio of 1:2. Thus, the aggregates retain their diameter and polydispersity index up to a 1:1 ratio, after which coagulation of particles begins.

Then, we studied the aggregation of systems **3a**–**d/5** and meroterpenoids **3a**–**d** in water by the DLS method (Table 2). Associates of geranyl and farnesyl derivatives **3a** and **3b** presumably have different structures. This is proven by insignificant changes in the average hydrodynamic diameter of aggregates in the case of geranylpyridinium salt, and significant changes in the average hydrodynamic diameter of aggregates in the case of farnesylpyridinium salt. Aggregates of phytylpyridinium salt **3c** have the same hydrodynamic diameters in the presence and in the absence of pillar[5]arene **5**. Apparently, this is due to the conformational softness of the phytyl chain, in which there are fewer double bonds, in contrast to geranyl chain and farnesyl chain.

We also measured the zeta-potential of meroterpenoids **3a**–**d** in the absence and in the presence of pillar[5]arene **5** in various ratios in water (Figure 4). The value of the zeta-potential indicates the stability of the colloidal system. Point 0.0 (Figure 4) on the abscissa axis corresponds to the value of the zeta-potential for the meroterpenoid solution in the absence of pillar[5]arene. The zeta-potential of the system begins to change sharply when pillar[5]arene is added to the meroterpenoid solution. The dependence reaches a plateau at a molar ratio of 1:1 and zeta-potential values of −50–−70 mV. Negative values of the zeta potential indicate that carboxylate fragments of the pillar[5]arene are located on the surface of the aggregates.

Then, the interaction between pillar[5]arene **5** and meroterpenoids **3a,b** was studied by UV-vis spectroscopy. A hypochromic effect at the macrocycle wavelength λ = 295 nm are observed for systems **3a/5** and **3b/5**. Spectrophotometric titration methods were used to determine the binding constants. The absorption spectra of the **3a,b/5** systems were recorded with the concentration of pillar[5]arene **5** (1 × 10^−5^ M) remaining constant, and the various concentration of **3a,b** (1 × 10^−6^–1 × 10^−5^ M). The data obtained were processed by BindFit [27]. The association constant of **3a/5** (1:1) was 8557 M^−1^ and **3b/5** (1:1) was 14,900 M^−1^. Moreover, the stoichiometry of the complex was confirmed by titration data processed using host-guest ratios of 1:2 and 2:1. However, in this case, the association constants of the complexes were determined with a large error. That confirms the correctness of our chosen 1:1 model (see Appendix A). These data confirm the correctness of our chosen 1:1 stoichiometry.

Thus, ionic substitution occurs in aqueous solutions of derivatives **4a,b,d** in the presence of pillar[5]arene. Supramolecular amphiphiles **4a,b,d/5** are formed in a water-methanol mixture. In the case of phytol derivative **4c**, ion substitution occurs both in water and in a water-methanol mixture. It was also shown that the pyridinium fragment of the synthesized meroterpenoids **3a**–**d** are include into the pillar[5]arene cavity in water to form the supramolecular amphiphile **3a**–**d/5** according to the “host-guest” principle. BindFit calculations showed that the stoichiometry of inclusion complexes **3a,b/5** was 1:1.

### 2.3. Membranotropic Activity of Supramolecular Amphiphiles Based on Pillar[5]Arene and Meroterpenoids Containing Pyridinium and Imidazolium Fragments

Next, the interaction of supramolecular amphiphiles with model dipalmitoylphosphatidylcholine (DPPC) vesicles at pH = 7.4 was studied. We used the turbidimetry method to determine the temperature (T_m_) of the phase transition of DPPC vesicles. The binding of amphiphilic compounds by the lipid bilayer is accompanied by a change in the packing density of lipids. The phase transition temperature T_m_ (gel—liquid crystals) is a sensitive indicator of the state of lipid molecules in the bilayer. T_m_ was determined by measuring the turbidity of the water lipid dispersion with increasing temperature [28]. This method is convenient because it does not depend on the influence of scattering particles, because the phase transition is recorded as a sharp decrease in absorption within a certain narrow temperature range, typical of the selected lipid.

It was shown that pyridinium salt **3a** leads to a linear decrease in the phase transition temperature of lipid vesicles depending on the concentration of the substance (Figure 5a, red graph). Pillar[5]arene **5** does not lead to a change in the phase transition temperature of the vesicles. Similar studies for the system **3a/5** showed that in this case there is no noticeable decrease in the temperature (T_m_) of the phase transition of vesicles (Figure 5a, blue dots). The imidazolium derivative of geraniol **4a** leads to a linear decrease in the phase transition temperature of the vesicles (Figure 5b). However, the system **4a/5** does not lead to changes in the phase transition temperature (T_m_) of vesicles (Figure 5b). It means the absence of interaction of the supramolecular amphiphile **4a/5** with the model phospholipid membrane. The situation is similar for the derivatives of myrtenol—**3d** and **4d**. They lead to a linear decrease in the phase transition temperature (Figure 6, red graphs). Systems **3d/5** and **4d/5** does not lead to the decrease of phase transition temperature (T_m_) of the lipid vesicles (Figure 6, blue dots). It was also shown that meroterpenoids based on farnesol and phytol **3b,c** and **4b,c** lead to the solubilization of phospholipid vesicles above the molar ratio of substance/lipid of 0.1. However, supramolecular amphiphiles **3b/5**, **3c/5** and **4b/5** does not lead to changes in the phase transition temperature (T_m_) of lipid vesicles.

It can be concluded that meroterpenoids containing pyridinium fragments **3a,d** cause a greater decrease in temperature, in contrast to analogous imidazolium derivatives **4a**,**d**. Apparently, this is due to the higher polarity of the imidazolium fragment. It can also be concluded that acyclic monoterpenoid derivatives (with a geranyl fragment) **3a** and **4a** cause a greater decrease in the phase transition temperature (T_m_), in contrast to bicyclic monoterpenoids (with a myrtenyl fragment) **3d** and **4d**. This is most likely due to the size of the lipophilic substituent. The linear geranyl fragment can be more efficiently embedding into lipid bilayer, in contrast to the bicyclic myrtenyl fragment. Thus, based on the turbidimetric data, it follows that meroterpenoids **3a,d** and **4a,d** are embedded into lipid bilayers. At the same time, all systems of supramolecular amphiphiles **3a/5** and **4a/5** did not show any interaction with the phospholipid vesicle under these conditions, which makes it possible to use these associates as targeted drug delivery systems.

The lack of interaction between supramolecular amphiphiles and the phospholipid membrane is explained by the stability of the aggregates that form supramolecular amphiphiles. It has been suggested that a decrease in pH will disrupt the stability of the supramolecular amphiphiles aggregates and influence their interaction with the phospholipid membrane. The extracellular pH of tumor cells is known to be acidic [29,30]. Therefore, the phase transition temperatures of lipid vesicles in the presence of compound **3a** and the **3a**/**5** system at pH 4.1 were measured. The obtained dependence is presented in Figure 7. The supramolecular system of amphiphiles **3a/5** interacts with the phospholipid membrane at pH 4.1 (the blue graph on Figure 7). The linear nature of the dependence of the phase transition temperature (T_m_) of lipid vesicles indicates that the bilayer structure is not destroyed during interaction. Thus, it was shown that supramolecular amphiphiles interact with a lipid bilayer, only under certain conditions, with a possible pH-control.

### 2.4. Interaction of FUDR with Associates ***3a/5***

Further, to confirm the hypothesis about the possibility of using systems **3a**–**d**/**5** as components of DDS, we investigated the inclusion of the anticancer drug 5-fluoro-2′-deoxyuridine (floxuridine, FUDR) into the structure of the most thermodynamically stable associate (according to DLS data—Table 2) **3a/5** [31]. FUDR is used to treat colorectal, liver and stomach cancers. However, this drug is highly toxic, which causes a large number of side effects such as mouth ulcers, nausea, vomiting, hair loss, stomach ulcers, yellowing of the skin and eyes.

Inclusion of FUDR into associate **3a/5** was studied by DLS, ^19^F, 2D DOSY NMR spectroscopy methods. Thus, the study of ^19^F-NMR (Figure 8) spectra with proton decoupling FUDR and mixtures **3a/5/**FUDR in the ratios 1:1:1, 1:1:2, 1:1:5, 1:1:7, 1:1:10 (1 × 10^−3^ M) was carried out. To prepare studied solutions, FUDR was first dissolved in deionized water, then meroterpenoid **3a** was added. Then, the solution was thermostated at 25 °C for 10 min and the macrocycle **5** was added.

In the case of the inclusion of the FUDR molecule in the structure of the supramolecular associate, a shift of the fluorine signal to the weak field of the spectrum is observed [32]. It should be noted that in the case of the **3a/5/**FUDR system in a 1:1:1 ratio (1 × 10^−3^ M), a shift of the fluorine signal by Δδ ~1.30 ppm is observed (Figure 8) compared to free FUDR. Based on this, we can conclude that the shift of the fluorine signal in **3a/5/**FUDR to the downfield region of the spectrum indicates the inclusion of the FUDR molecule in the structure of the supramolecular associate, as well as the de-shielding of the fluorine atom caused by the presence of fluorine inside the associate **3a/5** [33]. An increase of the concentration of FUDR in the **3a/5/**FUDR system (1 × 10^−3^–1 × 10^−2^ M) leads to the return of the ^19^F signal to its original value. This is apparently associated with saturation of the **3a/5/**FUDR associates and an increase in the concentration of free FUDR. Also, the absence of interaction between FUDR and pillar[5]arene was established by ^1^H- and ^19^F-NMR spectroscopy.

DLS studies of the obtained aggregates **3a/5/**FUDR = 1: 1: 1 showed that associates are formed in the entire range of investigated concentrations (1 × 10^−3^–1 × 10^−5^ M). The most stable 3**a**/**5**/FUDR systems are formed at a concentration of 1 × 10^−4^ M. Aggregates with an average hydrodynamic diameter of 299 ± 10 nm and a PDI 0.25 are formed. The electrokinetic potential of the resulting systems was −78 ± 3 mV. In the case of destruction of the **3a/5** systems in the presence of FUDR, the zeta potential values would be closer to zero. Negative value of the zeta potential allows us to conclude that the FUDR is inside the **3a/5** associates.

The formation of **3a/5/**FUDR associates was additionally confirmed by 2D DOSY spectroscopy. Diffusion coefficients of FUDR, **3a/5** and **3a/5/**FUDR at 298 K (1 × 10^−3^ M) were determined. The DOSY spectrum of the **3a/5/**FUDR system in a 1:1:1 ratio shows the presence of associate signals lying on one straight line (Figure 9), with one diffusion coefficient (D = 4.01 × 10^−10^ m^2^s^−1^). This is significantly lower than the self-diffusion coefficient of FUDR (D = 7.65 × 10^−10^ m^2^s^−1^) and associates **3a/5** (D = 5.12 × 10^−10^ m^2^s^−1^) under the same conditions. The results obtained unambiguously indicate the incorporation of FUDR into the structure of the supramolecular associate **3a/5**. The results obtained are in good agreement with the literature data [34].

Thus, a stable self-assembling system of supramolecular amphiphiles based on pillar[5]arene **5** and meroterpenoids **3a–d** was obtained. Systems are able to form associates with FUDR. It was confirmed that FUDR is inside the associate **5**/**3a–d**. Using a pH-control, it is possible to release FUDR from associates in a targeted manner.

## 3. Experimental

### 3.1. General Information

All reagents and solvents (Sigma-Aldrich, St. Louis, MO, USA) were used directly as purchased or purified according to the standard procedures. The ^1^H-, ^13^C- and ^19^F-NMR spectra were recorded on an Avance 400 spectrometer (Bruker Corp., Billerica, MA, USA) (400 MHz for H-atoms) for 3–5% solutions in CDCl_3_, D_2_O, CD_3_OD, DMSO-d_6_. The residual solvent peaks were used as an internal standard. Elemental analysis was performed on a Perkin-Elmer 2400 Series II instrument (Perkin Elmer, Waltham, MA, USA). The FTIR ATR spectra were recorded on the Spectrum 400 FT-IR spectrometer (Perkin Elmer, Seer Green, Lantrisant, UK) with a Diamond KRS-5 attenuated total internal reflectance attachment (resolution 0.5 cm^−1^, accumulation of 64 scans, recording time 16 s in the wavelength range 400–4000 cm^−1^). HRMS mass spectra were obtained on a quadrupole time-of-flight (t, qTOF) AB Sciex Triple TOF 5600 mass spectrometer (AB SCIEX PTE. Ltd., Singapore) using a turbo-ion spray source (nebulizer gas nitrogen, a positive ionization polarity, needle voltage 5500 V). Recording of the spectra was performed in “TOF MS” mode with collision energy 10 eV, de-clustering potentially 100 eV and with a resolution of more than 30,000 full-width half-maximum. Samples with the analyte concentration of 5 µmol/L were prepared by dissolving the test compounds in the mixture of methanol (HPLC-UV Grade, Darmstadt, Germany). Pillar[5]arene **5** was synthesized according to the literature method [25].

### 3.2. General Procedure for the Synthesis of Compounds ***2a–d***

A solution of 0.02 mol of the appropriate terpene alcohol (geraniol, farnesol, or phytol) and 0.021 mol of DIPEA in 60 mL chloroform was prepared in a 250-mL round-bottom flask equipped with a Claisen adapter. The resulting solution was cooled to −5 °C. The Claisen adapter was fitted with a thermometer and a dropping funnel containing a solution of 0.021 mol of bromoacetyl bromide in 10 mL of chloroform. This solution was added at such a rate that the temperature did not rise above −2 °C. After the addition, the reaction mixture was left for 1 h at room temperature. The resulting reaction mixture was then washed with 5% aqueous Na_2_CO_3_ solution (2 × 50 mL), then washed with 50 mL of water. The solvent was removed on a rotary evaporator, after which the crude product was purified by column chromatography on silica gel (eluent hexane-propanol-2 20:1).

#### 3.2.1. Geranyl-2-bromoacetate (**2a**)

Pale yellow oil, yield: 4.29 g (78%). $n_{D}^{20}$ = 1.4895. ^1^H-NMR (CDCl_3_, *δ*, ppm, *J*/Hz): 1.60 (s, 3H, CH_3_), 1.68 (s, 3H, CH_3_), 1.71 (s, 3H, CH_3_), 2.04–2.11 (m, 4H, CH_2_–CH_2_), 3.83 (s, 2H, O=C–CH_2_Br), 4.68 (d, 2H, =CH–CH_2_–O, *^3^J_HH_* = 7.2), 5.07 (m, 1H, =CH), 5.35 (m, 1H, =CH). ^13^C-NMR (CDCl_3_, *δ*, ppm): 16.54, 17.71, 25.70, 26.11, 26.22, 39.52, 63.13, 117.36, 123.61, 131.93, 143.58, 167.23. FTIR ATR (*ν*, cm^−1^): 2967 (CH_2_Br); 2915 (CH_2_); 2856 (CH_3_); 1735 (C=O); 1444, 1424, 1408 (=CH); 1376 (CH_3_); 1276 (C–O–C); 1204, 1153 (C–O–C); 953, 888, 830 (=CH); 554 (CH_2_Br). Elemental analysis. Calculated for C_12_H_19_BrO_2_: C, 52.38; H, 6.96; Br, 29.04. Found: C, 52.13; H, 6.77; Br, 28.90.

#### 3.2.2. Farnesyl-2-bromoacetate (**2b**)

Pale yellow oil, yield: 5.15 g (75%). $n_{D}^{20}$ = 1.4970. ^1^H-NMR (CDCl_3_, *δ*, ppm, *J*/Hz): 1.60 (s, 6H, CH_3_), 1.68 (s, 3H, CH_3_), 1.72 (s, 3H, CH_3_), 1.96–2.15 (m, 8H, CH_2_–CH_2_), 3.84 (s, 2H, O=C–CH_2_Br), 4.69 (d, 2H, =CH–CH_2_–O, ^3^*J*_HH_ = 7.3), 5.09 (m, 2H, =CH), 5.35 (m, 1H, =CH). ^13^C-NMR (CDCl_3_, *δ*, ppm): 16.04, 16.56, 17.71, 25.72, 26.03, 26.12, 26.58, 26.71, 39.53, 39.70, 39.81, 63.15, 117.36, 123.49, 124.29, 131.37, 135.58, 135.71, 143.62, 167.25. FTIR ATR (*ν*, cm^−1^): 2965 (CH_2_Br); 2922 (CH_2_); 2856 (CH_3_); 1735 (C=O); 1669 (C=C), 1445 (=CH); 1376, 1342 (CH_3_); 1276 (C–O–C); 1206, 1154 (C–O–C); 956, 887, 828 (=CH); 553 (CH_2_Br). Elemental analysis. Calculated for C_17_H_27_BrO_2_. C, 59.48; H, 7.93; Br, 23.27. Found: C, 59.43; H, 7.97; Br, 22.98.

#### 3.2.3. Phytyl-2-bromoacetate (**2c**)

Pale yellow oil, yield: 5.76 g (69%). $n_{D}^{20}$ = 1.4710. ^1^H-NMR (CDCl_3_, *δ*, ppm, *J*/Hz): 0.83 (s, 3H, CH_3_), 0.85 (br. s,6H, CH_3_), 0.87 (s, 3H, CH_3_), 1.04–1.10 (m, 4H, CH_2_–CH_2_), 1.12–1.16 (m, 2H, CH_2_), 1.21–1.26 (m, 6H, CH_2_–CH_2_), 1.28–1.32 (m, 2H, CH_2_) 1.34–1.42 (m, 4H, CH_2_–CH_2_), 1.48–1.56 (m, 1H, CH), 1.73 (br. s, 3H, CH_3_), 2.01 (m, 1H, CH), 2.07 (m, 1H, CH), 3.84 (br s, 2H, O=C–CH_2_Br), 4.67 (m, 2H, =CH–CH_2_–O), 5.35 (m, 1H, =CH). ^13^C-NMR (CDCl_3_, *δ*, ppm): 16.46, 19.62, 19.69, 19.76, 22.65, 22.74, 23.53, 24.47, 24.82, 25.00, 25.64, 26.06, 26.11, 27.99, 32.38, 32.68, 32.71, 32.72, 32.79, 36.60, 36.70, 36.81, 36.91, 37.31, 37.39, 37.44, 39.38, 39.86, 62.86, 63.19, 117.09, 117.85, 144.14, 144.60, 167.25. FTIR ATR (*ν*, cm^−1^): 2952 (CH_2_Br); 2925 (CH_2_); 2867 (CH_3_); 2847 (CH_2_); 1738 (C=O); 1461, 1423 (=CH); 1377, 1366 (CH_3_); 1276 (C–O–C); 1208, 1154 (C–O–C); 955, 886, 735 (=CH); 555 (CH_2_Br). Elemental analysis. Calculated for C_22_H_41_BrO_2_. C, 63.30; H, 9.90; Br, 19.14. Found: C, 64.13; H, 9.97; Br, 18.98.

#### 3.2.4. R-Myrtenyl-2-bromoacetate (**2d**)

Pale yellow oil, yield: 3.88 g (71%). $n_{D}^{20}$ = 1.5698. ^1^H-NMR (CDCl_3_, *δ*, ppm, *J*/Hz): 0.82 (s, 3H, CH_3_), 1.18 (d, 1H, CH, ^3^*J*_HH_ = 8.7), 1.29 (s, 3H, CH_3_), 2.07–2.17 (m, 2H, CH_2_), 2.21–2.36 (m, 2H, CH_2_), 2.41 (m, 1H, CH), 3.83 (s, 2H, O=C–CH_2_Br), 4.55 (s, 2H, =CH–CH_2_–O), 5.62 (s, 1H, =CH). ^13^C-NMR (CDCl_3_, *δ*, ppm): 21.14, 26.00, 26.14, 31.34, 31.52, 38.10, 40.64, 43.54, 68.79, 122.74, 142.25, 167.23. FTIR ATR (*ν*, cm^−1^): 2986 (CH_2_Br); 2916 (CH_2_); 2832 (CH_3_); 1735 (C=O); 1469, 1447, 1429(=CH); 1366 (CH_3_); 1272 (C–O–C); 1216, 1204, 1160, 1129 (C–O–C); 1044 (C–O–C); 1007, 963 (cyclobutane moiety vibrations); 887, 801 (=CH); 780 (=CH); 547 (CH_2_Br). Elemental analysis. Calculated for C_12_H_17_BrO_2_. C, 52.76; H, 6.27; Br, 29.25. Found: C, 53.17; H, 6.61; Br, 27.74;

### 3.3. General Procedure for the Synthesis of Compounds ***3a–d*** and ***4a–d***

In a 25 mL round-bottom flask, a solution of 1.5 mmol of terpenyl bromoacetate (**2a**–**d**) in 12 mL of diethyl ether was prepared. To the resulting solution 1.6 mmol of pyridine or *N*-methylimidazole was added. The reaction mixture was stirred at room temperature (in the case of **3c**, **4b,c** heating under reflux was required) for 36 h. Then diethyl ether was removed on a rotary evaporator. The crude residue was washed with hexane to get rid of the starting reagents, then dried on a rotary evaporator under vacuum.

#### 3.3.1. 1-(2-(Geranyloxy)-2-oxoethyl)pyridinium bromide (**3a**)

Brown resin, yield: 0.46 g (87%). ^1^H-NMR (CDCl_3_, *δ*, ppm, *J*/Hz): 1.59 (s, 3H, CH_3_), 1.67 (s, 3H, CH_3_), 1.70 (s, 3H, CH_3_), 2.03–2.10 (m, 4H, CH_2_–CH_2_), 4.73 (d, 2H, =CH–CH_2_–O, ^3^*J*_HH_ = 7.3), 5.06 (m, 1H, =CH), 5.32 (m, 1H, =CH), 6.24 (s, 2H, O=C–CH_2_N^+^), 8.07 (t, 2H, N^+^=CH_ar._–CH_ar._, ^3^*J*_HH_ = 7.4, 6.9), 8.52 (t, 1H, N^+^=CH_ar._–CH_ar._=CH_ar__._, ^3^*J*_HH_ = 7.8), 9.38 (d, 2H, N^+^=CH_ar._, ^3^*J*_HH_ = 6.0). ^13^C-NMR (CDCl_3_, *δ*, ppm.): 16.55, 17.74, 25.74, 26.21, 39.52, 57.91 63.13, 117.36, 123.61, 125.26, 131.93, 140.44, 143.58, 145.51, 171.12. FTIR ATR (*ν*, cm^−1^): 2972 (CH_3_); 2930, 2901 (CH_2_); 1740 (C=O); 1638 (C=C); 1377 (CH_3_); 1241, 1204 (Pyridine); 1075, 1066, 1058, 1028 (C–O–C); 892(C=C); 775 (CH_ar._). Elemental analysis. Calculated for C_17_H_24_BrNO_2_. C, 56.99; H, 6.69; Br, 22.36; N, 4.01. Found: C, 57.63; H, 6.83; Br, 22.55; N, 3.95.

#### 3.3.2. 1-(2-(Farnesyloxy)-2-oxoethyl)pyridinium bromide (**3b**)

Brown resin, yield: 0.53 g (84%). ^1^H-NMR (CDCl_3_, *δ*, ppm, *J*/Hz): 1.59 (s, 6H, CH_3_), 1.68 (s, 6H, CH_3_), 1.92–2.13 (m, 8H, CH_2_–CH_2_), 4.74 (d, 2H, =CH–CH_2_–O, ^3^*J*_HH_ = 7.4), 5.08 (m, 2H, =CH), 5.34 (m, 1H, =CH), 6.24 (s, 2H, O=C–CH_2_N^+^), 8.07 (t, 2H, N^+^=CH_ar._–CH_ar._, ^3^*J*_HH_ = 7.0), 8.51 (t, 1H, N^+^=CH_ar._–CH_ar._=CH_ar__._, ^3^*J*_HH_ = 7.8), 9.36 (d, 2H, N^+^=CH_ar_, ^3^*J*_HH_ = 6.0). ^13^C-NMR (CDCl_3_, *δ*, ppm): 16.01, 16.51, 17.69, 25.74, 26.13, 26.14, 26.61, 39.53, 39.70, 39.81, 58.17, 63.15, 117.33, 123.51, 124.32, 125.29, 131.34, 135.62, 135.69, 140.11, 143.58, 145.49, 167.23. FTIR ATR (*ν*, cm^−1^): 2969 (CH_3_); 2927 (CH_2_); 1743, 1713 (C=O); 1637 (C=C); 1445(=CH); 1376 (CH_3_); 1200 (Pyridine); 1076, 1057, 1027 (C–O–C); 775 (CH_ar._). Elemental analysis. Calculated for C_22_H_32_BrNO_2_. C, 63.17; H, 7.72; Br, 18.32; N, 3.41. Found: C, 62.56; H, 7.64; Br, 18.92; N, 3.32.

#### 3.3.3. 1-(2-(Phytyloxy)-2-oxoethyl)pyridinium bromide (**3c**)

Pale brown resin, yield: 0.62 g (73%). ^1^H-NMR (CDCl_3_, *δ*, ppm, *J*/Hz): 0.83 (br. s, 3H, CH_3_), 0.85 (br. s, 6H, CH_3_), 0.87 (s, 3H, CH_3_), 1.04–1.10 (m, 4H, CH_2_–CH_2_), 1.11–1.14 (m, 2H, CH_2_), 1.20–1.25 (m, 6H, CH_2_–CH_2_), 1.26–1.30 (m, 2H, CH_2_) 1.34–1.42 (m, 4H, CH_2_–CH_2_), 1.47–1.55 (m, 1H, CH), 1.72 (br. s, 3H, CH_3_), 2.02–2.10 (m, 2H, CH), 4.73 (m, 2H, =CH–CH_2_–O), 5.32 (m, 1H, =CH), 6.28 (s, 2H, O=C–CH_2_N^+^), 8.06 (t, 2H, N^+^=CH_ar._–CH_ar._, ^3^*J*_HH_ = 7.0), 8.50 (t, 1H, N^+^=CH_ar._–CH_ar._=CH_ar__._, ^3^*J*_HH_ = 7.7), 9.39 (d, 2H, N^+^=CH_ar_, ^3^*J*_HH_ = 6.0). ^13^C-NMR (CDCl_3_, *δ*, ppm): 16.47, 19.67, 19.72, 19.78, 22.69, 22.75, 23.58, 24.50, 25.01, 25.02, 25.69, 26.12, 26.19, 32.40, 32.71, 32.73, 32.76, 32.80, 36.62, 36.72, 36.84, 36.93, 37.32, 37.41, 37.47, 39.41, 39.90, 57.91, 62.88, 63.21, 117.11, 117.91,125.77, 140.88, 144.20, 144.62, 145.90, 169.13. FTIR ATR (*ν*, cm^−1^): 2952 (CH_3_); 2924, 2901 (CH_2_); 1737 1734 (C=O); 1491 (=CH); 1378 (CH_3_); 1207 (Pyridine); 1195, 1058 (C–O–C); 777 (CH_ar._). HRMS: calculated [M − Br^−^]^+^ *m/z* = 416.3524, found: [M − Br^−^]^+^ *m/z* = 416.3487.

#### 3.3.4. 1-(2-(R-Myrtenyloxy)-2-oxoethyl)pyridinium bromide (**3d**)

Yellow resin, yield: 0.4 g (75%). ^1^H-NMR (CDCl_3_, *δ*, ppm, *J*/Hz): 0.77 (s, 3H, CH_3_), 1.13 (d, 1H, CH, ^3^*J*_HH_ = 8.6), 1.29 (s, 3H, CH_3_), 2.07–2.14 (m, 2H, CH_2_), 2.20–2.36 (m, 2H, CH_2_), 2.41 (m, 1H, CH), 4.58 (s, 2H, CH_2_–O), 5.62 (s, 1H, =CH), 6.25 (m, 2H, O=C–CH_2_N^+^), 8.07 (t, 2H, N^+^=CH_ar._–CH_ar._, ^3^*J*_HH_ = 6.9), 8.51 (t, 1H, N^+^=CH_ar._–CH_ar._=CH_ar__._, ^3^*J*_HH_ = 8.0), 9.37 (d, 2H, N^+^=CH_ar_, ^3^*J*_HH_ = 6.0). ^13^C-NMR (DMSO-d_6_, *δ*, ppm.): 21.50, 26.49, 32.24, 32.35, 38.95, 41.87, 44.66, 61.87, 70.38, 124.15, 129.11, 129.23, 143.38, 147.58, 148.08, 167.08. FTIR ATR (*ν*, cm^−1^): 3039, 3017 (=C–H); 2989 (CH_3_); 2916, 2882, 2830 (CH_2_); 1749 (C=O); 1638, 1502 (C=N_pyridine_); 1468 (CH_3_); 1444, 1430 (=C–H); 1368 (CH_3_); 1267 (C–O–C); 1243, 1160, 1131 (C–O–C); 950 (cyclobutane moiety vibrations); 785, 774 (=CH); 671 (=CH). HRMS: calculated [M − Br^−^]^+^ *m/z* = 272.1645, found: [M − Br^−^]^+^ *m/z* = 272.1651.

#### 3.3.5. 3-(2-(Geranyloxy)-2-oxoethyl)-1-methyl-1H-imidazole-3-ium bromide (**4a**)

Yellow resin, yield: 0.44 g (83%). ^1^H-NMR (CDCl_3_, *δ*, ppm, *J*/Hz): 1.59 (s, 3H, CH_3_), 1.67 (s, 3H, CH_3_), 1.70 (s, 3H, CH_3_), 2.03–2.10 (m, 4H, CH_2_–CH_2_), 4.07 (s, 3H, CH_3_N), 4.71 (d, 2H, =CH–CH_2_–O, ^3^*J*_HH_ = 7.3), 5.05 (m, 1H, =CH), 5.32 (m, 1H, =CH), 5.43 (s, 2H, O=C–CH_2_N^+^), 7.41 (s, 1H, CH_imidazole_), 7.52 (s, 1H, CH_imidazole_), 10.21 (s, 1H, CH_imidazole_). ^13^C-NMR (CDCl_3_, *δ*, ppm): 16.66, 17.80, 25.77, 26.27, 36.99, 39.60, 59.34, 63.73, 116.91, 123.12, 123.60, 123.77, 132.06, 138.45, 144.28, 166.19. FTIR ATR (*ν*, cm^−1^): 3401, 3154, 3098 =(=C–H); 2966 (CH_3_); 2915, 2856 (CH_2_); 1623 (C=O); (C=C); 1577 (C=N_imidazole_); 1437_._ (CH_2_CO); 1381, 1345 (CH_3_); 1215, 1196, 1173 (C–O–C); 1104 (C–O–C); 1035 (C=N_imidazole_); 975, 941, 892(=C–H); 775, 753, 756 (CH_imidazole_). Elemental analysis. Calculated for C_21_H_33_BrN_2_O_2_. C, 53.79; H, 7.05; Br, 22.36; N, 7.84. Found: C, 53.81; H, 7.09; Br, 22.29; N, 7.81.

#### 3.3.6. 3-(2-(Farnesyloxy)-2-oxoethyl)-1-methyl-1H-imidazole-3-ium bromide (**4b**)

Yellow resin, yield: 0.52 g (81%). ^1^H-NMR (CDCl_3_, *δ*, ppm, *J*/Hz): 1.60 (s, 6H, CH_3_), 1.68 (s, 3H, CH_3_), 1.72 (s, 3H, CH_3_), 1.95–2.11 (m, 8H, CH_2_–CH_2_), 4.08 (s, 3H, CH_3_N), 4.73 (d, 2H, =CH–CH_2_–O, ^3^*J*_HH_ = 7.5), 5.08 (t, 2H, =CH, ^3^*J*_HH_ = 6.4), 5.34 (t, 1H, =CH, ^3^*J*_HH_ = 6.9), 5.43 (s, 2H, O=C–CH_2_N^+^), 7.29 (s, 1H, CH_imidazole_), 7.42 (s, 1H, CH_imidazole_), 10.52 (s, 1H, CH_imidazole_). ^13^C-NMR (CDCl_3_, *δ*, ppm): 16.63, 19.30, 24.57, 27.07, 33.16, 37.12, 39.65, 50.17, 61.62, 119.87, 124.65, 131.65, 134.83, 140.55, 165.26. FTIR ATR (*ν*, cm^−1^): 2966 (CH_3_); 2915 (CH_2_); 1746 (C=O); 1437 (CH_2_CO); 1382, 1377, 1365 (CH_3_); 1271, 1215, 1197, 1173 (C–O–C); 973, 941(=C–H); 732, 703, 621 (CH_imidazole_). Elemental analysis. Calculated for C_21_H_33_BrN_2_O_2_. C, 59.29; H, 7.82; Br, 18.78; N, 6.59. Found: C, 59.30; H, 7.79; Br, 18.89; N, 6.52.

#### 3.3.7. 3-(2-(Phytoloxy)-2-oxoethyl)-1-methyl-1H-imidazole-3-ium bromide (**4c**)

Yellow resin, yield: 0.58 g (78%). ^1^H-NMR (CDCl_3_, *δ*, ppm, *J*/Hz): 0.82 (br. s,3H, CH_3_), 0.84 (br. s, 6H, CH_3_), 0.86 (s, 3H, CH_3_), 1.02–1.08 (m, 4H, CH_2_–CH_2_), 1.10–1.14 (m, 2H, CH_2_), 1.19–1.24 (m, 6H, CH_2_–CH_2_), 1.26–1.29 (m, 2H, CH_2_) 1.32–1.40 (m, 4H, CH_2_–CH_2_), 1.46–1.53 (m, 1H, CH), 1.72 (br. s,3H, CH_3_), 1.98–2.08 (m, 2H, CH), 4.08 (s, 3H, CH_3_N), 4.71 (m, 2H, =CH–CH_2_–O), 5.32 (m, 1H, =CH), 5.42 (s, 2H, O=C–CH_2_N^+^), 7.36 (s, 1H, CH_imidazole_), 7.47 (s, 1H, CH_imidazole_), 10.29 (s, 1H, CH_imidazole_). ^13^C-NMR (CDCl_3_, *δ*, ppm.): 15.19, 18.44, 18.48, 18.55, 21.43, 21.53, 22.26, 23.36, 23.63, 24.16, 26.54, 30.95, 31.21, 31.29, 35.21, 35.92, 37.93, 48.63, 49.11, 61.50, 61.84, 115.71, 116.67, 122.07, 122.83, 136.68, 142.70, 165.18, 166.67. FTIR ATR (*ν*, cm^−1^): 2952 (CH_3_); 2925, 2868 (CH_2_); 1747 (C=O); 1577, 1566 (C=N_imidazole_); 1462 (=CH); 1377, 1366 (CH_3_); 1218 (Imidazole); 1199, 1174 (C–O–C); 942, 621 (CH_imidazole_). HRMS: calculated [M − Br^−^]^+^ *m/z* = 419.3632, found [M − Br^−^]^+^ *m/z* = 419.3638.

#### 3.3.8. 3-(2-(R-Myrtenyloxy)-2-oxoethyl)-1-methyl-1H-imidazole-3-ium bromide (**4d**)

Yellow resin, yield: 0.45 g (84%). ^1^H-NMR (CDCl_3_, *δ*, ppm, *J*/Hz): 0.79 (s, 3H, CH_3_), 1.15 (d, 1H, CH, ^3^*J*_HH_ = 8.8), 1.29 (s, 3H, CH_3_), 2.10–2.15 (m, 2H, CH_2_), 2.22–2.34 (m, 2H, CH_2_), 2.41 (m, 1H, CH), 4.07 (s, 3H, CH_3_N), 4.57 (s, 2H, CH_2_–O), 5.43 (m, 2H, O=C–CH_2_N^+^), 5.62 (s, 1H, =CH), 7.35 (s, 1H, CH_imidazole_), 7.44 (s, 1H, CH_imidazole_), 10.30 (s, 1H, CH_imidazole_). ^13^C NMR (CDCl_3_, *δ*, ppm): 21.05, 26.02, 31.23, 31.43, 36.91, 37.99, 40.45, 43.39, 50.23, 69.33, 123.29, 123.36, 123.77, 138.12, 141.47, 165.97. FTIR ATR (*ν*, cm^−1^): 3409, 3147, 3067 (=C–H); 2986 (CH_3_); 2914, 2831 (CH_2_); 1747 (C=O); 1627, 1576, (C=N_imidazole_); 1467 (CH_3_); 1430(=C–H); 1382, 1341 (CH_3_); 1265 (C–O–C); 1171 (C–O–C); 952 (cyclobutane moiety vibrations); 795, 776 (=CH); 697_._(=CH). HRMS: calculated [M − Br^−^]^+^ *m/z* = 275.1754, found: [M − Br^−^]^+^ *m/z* = 275.1760.

### 3.4. Determination of the Hydrodynamic Particle Size by Dynamic Light Scattering

The particle size distribution was determined by dynamic light scattering on a Zetasizer Nano ZS nanoparticle size analyzer (Malvern, Worcestershire, UK) in quartz cuvettes. The instrument is equipped with a 4 mV He-Ne laser operating at 633 nm. The measurements were carried out at a measurement angle of 173°, and the measurement position inside the cuvettes was determined automatically. The results were processed using the DTS program (Dispersion Technology Software 4.20). To prepare solutions, deionized water with a resistance of 18.0 MΩ∙cm, obtained using a Millipore-Q purification system, or methanol (HPLC-UV Grade, Darmstadt, Germany) were used. During the experiment, the concentrations of compounds varied from 1 × 10^−5^ to 1 × 10^−3^ M.

### 3.5. Measurement of the Zeta-Potential

The electrokinetic potential of aggregates compounds was measured on a Zetasizer Nano ZS instrument at 25 °C. The device is equipped with a 4 mW He-Ne laser operating at 633 nm and includes non-invasive backscatter optics. The measurements were carried out at a detection angle of 173°, and the position of the measurement in the cuvette was automatically determined by the software. The results were processed using the DTS package (Dispersion Technology Software 4.20).

### 3.6. Turbidimetry

Experiments to determine the temperature of the phase transition were carried out by measuring the turbidity of a dilute lipid suspension (0.7 mM) on a UV-3600 spectrophotometer (Shimadzu, Kyoto, Japan) equipped with Peltier temperature control unit, the thickness of the transmission layer was 1 cm, the slit width was 1 nm, and the wavelength was 400 nm. Titration of the lipid by the test compounds was carried out in quartz cells. To reduce the experimental error, the obtained compounds were added to 3 mL of a lipid suspension (0.7 mmol) as their concentrated solutions into a buffer 7.4 (50 mM Tris-HCl, 150 mM NaCl, pH 7.4) or buffer 4.1 (50 mM AcOH, pH 4.1). To determine the temperature of the lipid phase transition, we measured the optical density of the samples in the temperature range 38–43 °C in steps of 0.1 °C per minute. Experimental data on the dependence of the optical density of the emulsion of the vesicles were mathematically processed in the software package Origin 8.1 (OriginLab Corporation, Northampton, MA, USA) by the Vant-Hoff 2-state model giving the phase transition temperature (T_m_) of the system [28].

### 3.7. 2D DOSY NMR Spectroscopy

^1^H diffusion ordered spectroscopy (DOSY) spectra were recorded on a Bruker Avance 400 spectrometer at 9.4 Tesla at a resonating frequency of 400.17 MHz for 1H using a BBO Bruker 5 mm gradient probe. The temperature was regulated at 298 K and no spinning was applied to the NMR tube. DOSY experiments were performed using the STE bipolar gradient pulse pair (stebpgp1s) pulse sequence with16 scans of 16 data points collected. The maximum gradient strength produced in the z direction was 5.35 Gmm^−1^. The duration of the magnetic field pulse gradients (δ) was optimized for each diffusion time (Δ) in order to obtain a 2% residual signal with the maximum gradient strength. The values of δ and Δ were 1.800 μs and 100 ms, respectively. The pulse gradients were incremented from 2 to 95% of the maximum gradient strength in a linear ramp [35].

## 4. Conclusions

For the first time and using an original methodology, meroterpenoids containing hydrophilic pyridine **3a**–**d** and imidazole fragments **4a**–**d**, as well as hydrophobic terpene residues of geraniol, myrtenol, farnesol, and phytol were synthesized. The ability of the water-soluble pillar[5]arene **5** containing carboxylate fragments to form supramolecular amphiphiles by the principle of the formation of host-guest complexes with synthesized meroterpenoids containing pyridine fragments **3a**–**d** was shown. The association constants of the complexes were determined by UV-vis spectroscopy: K_ass_ (**3a/5**) = 8557 M^−1^ and K_ass_ (**3b/5**) = 14900 M^−1^. Analysis of the binding isotherms obtained using the BindFit software showed that the stoichiometry of the **5**/meroterpenoid was 1:1. It was shown by dynamic light scattering that supramolecular amphiphiles **3a**–**d**/**5** form monodisperse associates in a wide concentration range (1 × 10^−3^–1 × 10^−5^ M). **3a**/**5** systems do not interact with the DPPC model cell membrane at pH 7.4, but the associate is destroyed with a change in pH (pH 4.1) according to turbidimetry data. Upon the destruction of the associate, the initial meroterpenoid **3a** is released and embedded into the DPPC lipid bilayer. Using the methods of dynamic light scattering, ^19^F, 2D DOSY NMR spectroscopy, the incorporation of the antitumor drug 5-fluoro-2′-deoxyuridine (floxuridine) into the structure of the supramolecular associate **3a**/**5** was shown. DLS studies of the obtained **3a**/**5**/Floxuridine aggregates with a composition of 1:1:1 (1 × 10^−3^M) showed that associates are formed in the entire range of investigated concentrations (1 × 10^−3^–1 × 10^−5^ M). The most stable 3**a/5**/Floxuridine systems are formed at a concentration of 1 × 10^−4^ M. Aggregates with an average hydrodynamic diameter of 299 ± 10 nm and a PDI of 0.25 are formed. The electrokinetic potential of the obtained systems was –78 ± 3 mV, which indicates that the floxuridine is inside the **3a**/**5**. The results obtained may find application in the creation of a new generation of non-toxic biomimetic delivery systems for anticancer drugs.

## Data Availability

The data presented in this study are available in Appendix A.

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
