# Peer review of "Supramolecular Amphiphiles Based on Pillar[5]arene and Meroterpenoids: Synthesis, Self-Association and Interaction with Floxuridine"

_ijms, 2021, doi:10.3390/ijms22157950_

Round 1

Reviewer 1 Report

The manuscript reports an interesting study on self-assembly of supramolecular amphiphiles composed of water soluble pillar[5]arene and meroterpenoids containing pyridinium or imidazolium rests. The structure of the inclusion complexes were investigated with use of 1H NMR and DLS methods. Next, their aggregation and interaction with DPPC vesicles were studied and finally, the interaction of Floxuridine (FUDR) with the aggregates were investigated using 19F NMR and DOSY 2D NMR experiments.

Although there is nothing fundamentally new in the concept in these results and the synthetic methods used are rather routine the work has been carefully done, the compounds are fully characterized and the results are clearly presented. I suggest that the paper paper should be published in International Journal of Molecular Sciences after consideration of the following minor points:

– Since the UV-vis titration spectra (Figure S45) show only small absorption changes would the fluorescence (ref. 25) or 1H NMR titration methods (K. Han,… Eur. J. Org. Chem. 2013, 2057 or X. Shu,. Org. Biomol. Chem. 2012,10, 3393) afford more precise values of the association constants?

– What is the suggested structure of the 1:1:1 3a / 5 / FUDR complex? What about the interaction of FUDR with pure pillar[5]arene 5?

– What does the last sentence in sec 2.4 mean: “The results obtained are in good agreement with the literature data [34]”? The ref. 34 deals with completely different subject though the authors use DOSY 2D NMR.

Author Response

The manuscript reports an interesting study on self-assembly of supramolecular amphiphiles composed of water soluble pillar[5]arene and meroterpenoids containing pyridinium or imidazolium rests. The structure of the inclusion complexes were investigated with use of 1H NMR and DLS methods. Next, their aggregation and interaction with DPPC vesicles were studied and finally, the interaction of Floxuridine (FUDR) with the aggregates were investigated using 19F NMR and DOSY 2D NMR experiments.

Although there is nothing fundamentally new in the concept in these results and the synthetic methods used are rather routine the work has been carefully done, the compounds are fully characterized and the results are clearly presented. I suggest that the paper paper should be published in International Journal of Molecular Sciences after consideration of the following minor points:

Answer:

Dear Reviewer! Thank you very much for carefully reading and reviewing our paper!

– Since the UV-vis titration spectra (Figure S45) show only small absorption changes would the fluorescence (ref. 25) or 1H NMR titration methods (K. Han,… Eur. J. Org. Chem. 2013, 2057 or X. Shu,. Org. Biomol. Chem. 2012,10, 3393) afford more precise values of the association constants?

Answer:

Unfortunately, in this case fluorescence or 1H NMR titration methods do not give more precise values of the association constants, owing to low fluorescence intensity of pillar[5]arene 5 and absence of fluorescence of synthesized meroterpenoids. 1H NMR titration method is not suitable for our system due to aggregation processes at high concentrations.

– What is the suggested structure of the 1:1:1 3a / 5 / FUDR complex? What about the interaction of FUDR with pure pillar[5]arene 5?

Answer:

We assume that the FUDR molecules are inside the associates of supramolecular amphiphile 3a / 5. This is confirmed by 19F NMR spectroscopy data. A schematic picture of this associate is presented in a graphical abstract. The absence of interaction between FUDR and pillar[5]arene was established by 1H and 19F NMR spectroscopy.

We have added of suggested structure of the 1:1:1 3a/5/FUDR associate to the manuscript (Figure 9b).

Figure 9. a) 2D DOSY NMR spectrum 3a/5/ FUDR (1 × 10-3 М) in D2O (400 MHz, 298K), b) schematic picture of the associate 3a/5/FUDR.

We have added sentence about interaction of FUDR with pure pillar[5]arene 5 to the manuscript: «Also, the absence of interaction between FUDR and pillar[5]arene was established by 1H and 19F NMR spectroscopy.»

– What does the last sentence in sec 2.4 mean: “The results obtained are in good agreement with the literature data [34]”? The ref. 34 deals with completely different subject though the authors use DOSY 2D NMR.

Answer:

In this case, we are talking about the formation of a complex, multicomponent supramolecular system 3a/5/FUDR. In the presented article [34], a supramolecular polymer system of a similar structure is formed with the participation of pillar[5]arene. The formation of the supramolecular system in [34] was confirmed by a set of methods, including 2D DOSY experiments. We analyzed the results [34] and our results, came to similar conclusions about the formation of supramolecular systems by the 2D NMR DOSY method, but in our case spherical particles are formed. Therefore, the study [34] was included in the list of references.

Reviewer 2 Report

The manuscript of Akhmedov and co-workers deals with supramolecular amphiphiles based on pillar[5]arene and meroterpenoids. The synthesis, self-assembly and interaction with floxuridine is reported in the manuscript using various techniques such as turbidimetry, UV-vis spectroscopy, zeta-potential, dynamic light scattering, 19F and 2D DOSY NMR spectroscopy. The manuscript is globally well written and the topic is very interesting. The experimental section is well described as well as the products characterization. Therefore, the paper appropriated for publication in the International Journal of Molecular Sciences. However, some minor revisions are needed before publication.

1) I think that the value of binding constants obtained from UV-vis spectroscopy are too precise! For instance, “Kass (3a/5) = 8557.29 М-1 and Kass (3b/5) = 14900.25 М-1”, the “.29” or “.25” are clearly no sense: in your Figures S43 and S45, the S.D. are of ±10% of the given values! Please give the S.D. in the manuscript and correct your values… The mathematical treatment should not appear as is in the manuscript ...

2) Scheme 1: the charges are on the atoms as well as some the O of the C=O. Please correct.

3) Some typographic errors must be corrected. For instance, “N-methylimidazole” (page 13, line 401) is “N-methylimidazole”.

In conclusion, this manuscript must be revised before publication. However, after appropriate revision, this manuscript is appropriate for the International Journal of Molecular Sciences.

Author Response

The manuscript of Akhmedov and co-workers deals with supramolecular amphiphiles based on pillar[5]arene and meroterpenoids. The synthesis, self-assembly and interaction with floxuridine is reported in the manuscript using various techniques such as turbidimetry, UV-vis spectroscopy, zeta-potential, dynamic light scattering, 19F and 2D DOSY NMR spectroscopy. The manuscript is globally well written and the topic is very interesting. The experimental section is well described as well as the products characterization. Therefore, the paper appropriated for publication in the International Journal of Molecular Sciences. However, some minor revisions are needed before publication.

Answer:

Dear Reviewer! Thank you very much for carefully reading and reviewing our paper!

1) I think that the value of binding constants obtained from UV-vis spectroscopy are too precise! For instance, “Kass (3a/5) = 8557.29 М-1 and Kass (3b/5) = 14900.25 М-1”, the “.29” or “.25” are clearly no sense: in your Figures S43 and S45, the S.D. are of ±10% of the given values! Please give the S.D. in the manuscript and correct your values… The mathematical treatment should not appear as is in the manuscript ...

Answer:

According to the reviewer's comments, we have rounded the constants’ values to integers in manuscript.

2) Scheme 1: the charges are on the atoms as well as some the O of the C=O. Please correct.

Answer:

We have corrected Scheme 1 in the manuscript.

3) Some typographic errors must be corrected. For instance, “N-methylimidazole” (page 13, line 401) is “N-methylimidazole”.

Answer:

Reviewer's comments were taken into account. Corrections have added to the revised version of the manuscript.